# An evaluation of the early impact of the COVID-19 pandemic on Zambia's routine immunization program

**Amy K. Winter**[1][*], **Saki Takahashi**[2], **Andrea C. Carcelen**[3], **Kyla Hayford**[3], **Wilbroad Mutale**[4], **Francis D. Mwansa**[5], **Nyambe Sinyange**[6], **David Ngula**[5], **William J. Moss**[2,3], **Simon Mutembo**[3]

1 Department of Epidemiology and Biostatistics, University of Georgia, Athens, Georgia, United States of America, 2 Department of Epidemiology, Johns Hopkins Bloomberg School of Public Health, Baltimore, Maryland, United States of America, 3 Department of International Health, International Vaccine Access Center, Johns Hopkins Bloomberg School of Public Health, Baltimore, Maryland, United States of America, 4 School of Public Health, University of Zambia, Lusaka, Zambia, 5 Child Health Unit, Directorate of Public Health and Research, Ministry of Health, Lusaka, Zambia, 6 Field Epidemiology Training Program, Zambia National Public Health Institute, Lusaka, Zambia

☯ These authors contributed equally to this work.
* awinter@uga.edu

**Data Availability Statement:** Study protocol and dataset can be made available upon request to Godfrey Biemba, director Zambia National Health Research authority (gbiemba@gmail.com). Data

## Abstract

Implications of the COVID-19 pandemic for both populations and healthcare systems are vast. In addition to morbidity and mortality from COVID-19, the pandemic also disrupted local health systems, including reductions or delays in routine vaccination services and catch-up vaccination campaigns. These disruptions could lead to outbreaks of other infectious diseases that result in an additional burden of disease and strain on the healthcare system. We evaluated the impact of the COVID-19 pandemic on Zambia's routine childhood immunization program in 2020 using multiple sources of data. We relied on administrative vaccination data and Zambia's 2018 Demographic and Health Survey to project national disruptions to district-specific routine childhood vaccination coverage within the pandemic year 2020. Next, we leveraged a 2016 population-based serological survey to predict age-specific measles seroprevalence and assessed the impact of changes in vaccination coverage on measles outbreak risk in each district. We found minor disruptions to routine administration of measles-rubella and pentavalent vaccines in 2020. This was in part due to Zambia's Child Health Week held in June of 2020 which helped to reach children missed during the first six months of the year. We estimated that the two-month delay in a measles-rubella vaccination campaign, originally planned for September of 2020 but conducted in November of 2020 as a result of the pandemic, had little impact on modeled district-specific measles outbreak risks. This study estimated minimal increases in the number of children missed by vaccination services in Zambia during 2020. However, the ongoing SARS-CoV-2 transmission since our analysis concluded means efforts to maintain routine immunization services and minimize the risk of measles outbreaks will continue to be critical. The methodological framework developed in this analysis relied on routinely collected data to estimate disruptions of the COVID-19 pandemic to national routine vaccination program performance and

were obtained under data sharing agreements from Zambia Ministry of Health and the Zambia National Health Research Authority and will only be shared with permission from the Ministry of Health.

**Funding:** This work was financially supported by the Bill and Melinda Gates Foundation (https://www.gatesfoundation.org) in the form of a grant (OPP1094816) awarded to AKW, ACC, KH, WJM, and SM. No additional external funding was received for this study. The funder had no role in study design, data collection and analysis, decision to publish, or preparation of the manuscript.

**Competing interests:** The authors have declared that no competing interests exist.

its impact on children missed at the subnational level can be deployed in other countries or for other vaccines.

## Introduction

As of December 23, 2022 SARS-CoV-2 has been responsible for 652 million confirmed cases of COVID-19 globally [1]. Implications of the pandemic for both the population and the healthcare system are vast. In addition to direct morbidity and mortality from COVID-19, the pandemic also has the potential to cause disruptions to local health systems. Disruptions of particular concern are to routine immunization programs used to control the spread of other infectious diseases [2]. Disruptions to immunization programs could lead to outbreaks of vaccine-preventable diseases [3]. Reasons for disruptions include supply-side issues including international and domestic supply chain disruptions, border closures and trade restrictions, reduced capacity and health care service offerings, restrictions on movement, and assignment of vaccination staff to COVID-19 control activities [4–7]. Additionally, demand-side related issues may also negatively impact uptake of vaccines such as parental reluctance to seek vaccinations for their children because they do not want to risk being exposed to SARS-CoV-2 at health facilities.

As a result of the pandemic and the measures implemented to control the spread of SARS-CoV-2, routine immunization programs underperformed [2, 8]. According to the World Health Organization (WHO) the suspension or disruption of vaccination services in over 68 countries during the early stages of the pandemic put over 80 million infants younger than one year of age at risk of vaccine preventable diseases [6]. While countries are working hard to maintain optimal vaccination program performance, we have started to see the result of some of these disruptions. For example, in 2022 after years of having eliminated wild poliovirus, Malawi and Mozambique reported an outbreak of wild poliovirus and wild poliovirus cases have popped up in the United Kingdom and the United States of America [9]. Additionally, the World Health Organization has warned of "perfect storm" condition for measles outbreaks given measles high transmissibility, pandemic-related disruptions, and displaced populations due to conflicts or crises [10].

We evaluated the disruption of the COVID-19 pandemic on Zambia's national vaccination program and its impact on the number of missed vaccinations at the sub-national level. As it is difficult to establish a causal link between the pandemic and vaccination programs, we compared pre-pandemic and pandemic vaccination program performance and attributed the difference to pandemic-related disruptions. The time of assessment was January 2020 –October 2020, which encompassed one wave of COVID-19 cases in July 2020. Findings from this analysis were used by the Zambian Ministry of Health to inform country-specific 2020 vaccination strategic responses in light of the COVID-19 pandemic.

## Methods

We focused our analysis on the administration of routine bivalent measles-rubella vaccine dose 1 (MR1) and the pentavalent diphtheria, pertussis, tetanus toxoid, hepatitis B and *Haemophilus influenzae* type b vaccine doses 1 and 3 (Penta1, Penta3). In Zambia, the recommended childhood vaccination schedule includes three Penta doses at 6, 10, and 14 weeks of life and two MR doses at 9 and 18 months of age. We evaluated disruptions to measles-containing vaccines because clusters of few susceptible individuals pose a high risk of measles outbreak. We

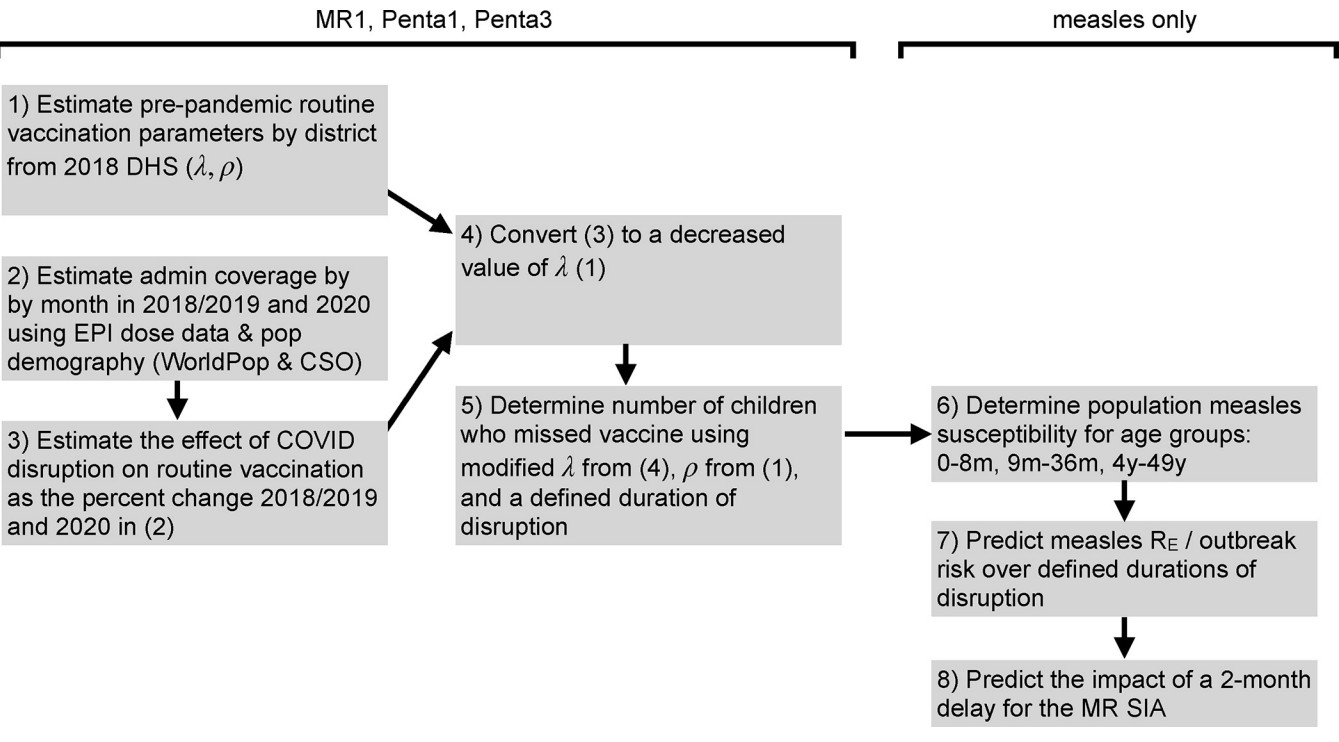

**Fig 1. Methodological roadmap.**

additionally evaluated disruptions of the pentavalent vaccine given that coverage of pentavalent vaccines are used as an indicator of immunization program performance [11, 12].

To understand if and how COVID-19 related health care disruptions impacted the risk of vaccine-preventable diseases, we estimated changes in the national routine immunization program during the COVID-19 pandemic year 2020, and its impact on the number of children who missed vaccination (and risk of measles outbreak) at the district level (administrative level 2). This included first estimating a pre-pandemic routine vaccination coverage rate at the district level from a population-based representative survey. We then evaluated the changes in routine vaccination in pandemic year 2020 by estimating a percent reduction in the rate of vaccination using administrative data. Reported vaccination coverage for MR1 and Penta1 had stabilized prior to the pandemic such it is reasonable to attribute changes to vaccination coverage in pandemic-year 2020 to the pandemic; Penta3 showed slight declines in 2018 and 2019 leading up the pandemic (S1 Fig). Finally, we calculated the number of children missed for each month of disruption. For measles, given the particularly high transmissibility among susceptible individuals, we took one step further and explored measles outbreak risk with and without vaccine disruptions, and evaluated the impact of a two-month delay of the national MR vaccination campaign. See Fig 1 for a methodological roadmap, and the following sub-sections detailing each analytic step.

## District-specific pre-pandemic routine vaccination coverage

We relied on childhood vaccination coverage data from Zambia's 2018 cross-sectional Demographic and Health Survey (DHS) made publicly available by ICF International to estimate routine vaccination coverage by district in Zambia. For each child, we extracted the following data from the DHS: age at the time of survey, whether the child had ever received an MR1,

Penta1, and Penta3 vaccine dose (based on vaccination card or report of the parent or guardian), and age at the time of vaccination (if a vaccination card was available). Data were available for 112 of the 116 designated districts in 2018 and for 5,670 children between the ages of 0 and 36 months. We conducted a modified survival analysis accounting for uncensored (i.e. child with vaccination date on vaccination card), left censored (i.e., mother reported vaccination but date of vaccination unknown), and right censored (i.e., child unvaccinated at the time of the DHS survey) data on the age of vaccination, extending a statistical approach previously described [13]. The probability that an individual was vaccinated with each vaccine by a given age depends on two estimated parameters: district-specific lifetime probability of being vaccinated through routine vaccination services ($\rho_i$) and the district-specific rate of receiving the vaccine through routine vaccination ($\lambda_i$) (S2 Fig); the latter accounts for differences in the timeliness of receiving routine vaccination as a function of age. To estimate these district-specific parameters, we assumed they were multivariate normally distributed with a district-specific mean that was specified by a conditional autoregressive model. We extended the granularity of DHS provincially representative data using spatial models that account for the data structure. As a result, district estimates depended on assumptions inherent in the conditional autoregressive model. See S1 Text for more details.

## National reduction in the pandemic year 2020 vaccination rate (compared to pre-pandemic years 2018 & 2019)

Administrative data on the number of vaccine doses delivered per month in each district from January 2018 to October 2020 for MR1, Penta1, and Penta3 was obtained from the Zambia Expanded Programme on Immunization (EPI). These data were used to estimate changes in the rate of routine vaccination in pandemic year 2020. Administrative vaccination data is generated by district health officers each month who aggregate health facility reports of the number of administered vaccine doses based on clinic records including registries.

We estimated national disruptions in the rate of routine vaccination because the administrative vaccination data were not sufficient to evaluate sub-national disruptions to routine vaccination (S3–S5 Figs). Given the seasonal nature of vaccine delivery due to Child Health Weeks (S6 Fig), we compared the vaccination coverage each month in 2018 and 2019 to 2020. We selected 2018 and 2019 as the baseline years for consistency with analysis on pre-pandemic vaccination coverage (i.e., DHS data). Note, we did not include administrative data in our estimate our pre-pandemic vaccination coverage (see the previous section) because these data are considered more unreliable due to incomplete or inaccurate reporting of vaccination, mistakes compiling the data across administrative units, and inaccurate population denominator estimates [14]. Rather, we only used administrative data to evaluate time trends or the change in vaccination. If data collection and associated biases in administrative vaccination data and estimated birth cohort are consistent across years, then estimated rates of disruptions would be robust to biases within the data.

We fit a binomial model to the 2018/2019 data and to the 2020 data, $N_{it} \sim Binomial\ (B_i, \gamma_t)$, where $N_{it}$ is the number of doses administered in district $i$ and month $t$, $B_i$ is the size of the birth cohort in district $i$, and $\gamma_t$ is the estimated proportion of individuals vaccinated in month $t$. We estimated the size of the birth cohorts in 2018/2019 and 2020 for each district by $B_i = P_i b_k$, where $P_i$ is the size of the population in the district $i$ estimated by aggregating WorldPop annual population estimates in 10x10 km grid cells over district polygons for the respective years [15, 16], and $b_k$ is the proportion of the population who are newborns (age 0) for the respective province $k$ in which district $i$ is located based on Zambia's central statistical office projections.

We estimated the disruption per month as the percent reduction in 2020 vaccination rate (compared to 2018/2019 vaccination rate) based on 2,000 draws of the 2020 posterior distribution of $\gamma_t$ and 2,000 draws of the 2018/2019 posterior distribution of $\gamma_t$. We calculated the mean and 95% uncertainty intervals of the precent reduction each month $t$. Given monthly variation, we then calculated the mean across 10 months (January 2020 –October 2020) to generate a national mean and 95% uncertainty interval of the percent reduction in routine rate for each vaccination dose, and assumed it was constant into the future.

## District-specific number of children missed by vaccination

We directly applied the national mean percent reduction in the rate of routine vaccination to the estimated district-specific pre-pandemic rate of routine vaccinations ($\lambda_i$) to estimate a modified district-specific proportion vaccinated over age in months in 2020. The duration of disruption determined the duration of time individuals were exposed to the reduced rate of vaccination compared to the baseline rate of vaccination. Therefore, for each month of disruption, we estimated a district and age-specific proportion vaccinated between 9 and 36 months of age. The number of children not vaccinated in each district was calculated as the sum of the number of individuals 9 to 36 months old times the proportion unvaccinated between 9 and 36 months (i.e., 1—proportion vaccinated). Individuals who aged out of this age group and remained unvaccinated were not included in these totals. These estimates were aggregated across districts to obtain national estimates.

## District-specific measles outbreak risk

We evaluated the district-specific additional risk of a measles outbreak for each month of disruption in 2020 and the impact of a national delay in the vaccination campaign from September to November 2020. We focused on measles given its high transmissibility and herd immunity threshold. To estimate outbreak risk, we first needed to estimate age-specific susceptibility. Population susceptibility was determined separately for three age bins: birth to 9 months, 9 months to 36 months (i.e., the age range of individuals with relevant measles vaccination data in the DHS survey), and 36 months to 49 years. We assumed that all infants from birth to nine months old had a level of protection by maternally derived antibodies beginning with 100% at birth and dropping by a rate of 0.45 until 8 months of age, and are susceptible thereafter until vaccinated [17]. We assumed no immunity from natural infection given the small number of measles cases reported since 2016 (average of 11 annual cases between 2016– 2019) [18].

We estimated susceptibility of individuals between 9 months and 36 months old by applying an age-specific vaccine effectiveness rate to the disruption modified district-specific probability an individual was vaccinated by age in months (as estimated above). We assumed all changes in the proportion susceptible by month of disruption was constrained to the 9- to 36-month-old age group who were eligible for MR1 and receiving it at a reduced rate, thereby ignoring the potential role that natural infection may have to reduce the impact of susceptibility on individuals younger than 9 months or older than 36 months. This assumption is justified by the lack of major measles outbreaks in 2020 in Zambia. The febrile-rash surveillance system in Zambia reported only 69 measles cases in 2020 to the World Health Organization, similar to the previous six years [19].

To estimate susceptibility among individuals older than 36 months, we relied on a hierarchical spatial model fit to measles seroprevalence data collected in 2016 [20]. Serological data provides the most direct estimate of measles immunity, obtained through vaccination or natural infection [21]. The measles IgG 2016 serological data came from a nested serosurvey within

the Zambia Population-Based HIV Impact Assessment consisting of 9,852 blood samples collected from individuals one month to 49 years old across all 72 districts at the time [20]. The model relied on routinely collected epidemiological data (i.e., vaccination coverage, suspected case data) and demographic data (i.e., age, district, province) to explain the variation in the cross-sectional 2016 seroprevalence data. To project measles seroprevalence in subsequent years, selected covariates from those years were combined with posterior estimates of model parameters. We did not have data on individuals over 49 years of age. We do not have estimates on susceptibility among populations over 49 years of age. However, given these birth cohorts were children when measles virus transmission was endemic in Zambia, it is reasonable to assume they would have been naturally exposed to measles virus and would therefore not be susceptible [22]. See S2 Text, and S7–S10 Figs for more details.

Lastly, to estimate outbreak risk, we calculated the measles effective reproductive number ($R_e$, i.e., the average number of individuals an infectious individual will infect in a partially susceptible population) for each month and district taking into account age-specific susceptibility and age-assortative mixing patterns [23]. See S3 Text for more details.

### Ethics statement

For the 2016 measles seroprevalence data, participants provided written informed consent, parental permission was obtained children under 18 years old, and assent was obtained for participants 10–17 years old. This serosurvey was conducted in accordance with relevant guidelines and regulations. Ethical approvals for protocols were provided by Johns Hopkins Bloomberg School of Public Health (00008423) as well as the Tropical Disease Research Center and the National Health Regulatory Agency in Zambia (TDRC/C4/01/2019).

## Results

### District-specific pre-pandemic routine vaccination coverage

In our pre-pandemic estimates of routine vaccination coverage using the 2018 Zambia DHS, we identified variation in district-level estimates of the lifetime probability and monthly rate of being vaccinated across age (Fig 2). The median lifetime probability of receiving MR1, Penta1, and Penta3 across districts was 0.94 (range 0.67 to 0.98), 0.99 (range 0.81 to 1.00), and 0.95 (range 0.56 to 0.99), respectively (Fig 2A–2C). There was some consistency in the estimated lifetime probability of being vaccinated across the three vaccines, i.e., districts with relatively lower lifetime probability of vaccination with MR1 also had lower probabilities of vaccination with Penta1 and Penta3 vaccines compared to other districts (correlation MR1 and Penta1 = 0.834; correlation MR1 and Penta3 = 0.764; correlation Penta1 and Penta3 = 0.801). The median monthly rate of routine vaccination for MR1, Penta1, and Penta3 across districts was 0.51 (range 0.29 to 0.72), 0.65 (range 0.38 to 0.85), and 0.41 (range 0.26 to 0.58), respectively (Fig 2D–2F); this is equivalent to a median average age of vaccination (among those vaccinated) of 9.96 months (range 9.39 to 11.48), 1.95 months (range 1.58 to 3.04), and 4.64 months (range 3.92 to 6.10) (S11 Fig). Districts with a lower rate of vaccination for MR1 did not necessarily have a lower rate of vaccination for Penta1 and Penta3 (correlation between MR1 and Penta1 = 0.44, correlation between MR1 and Penta3 = 0.46), although the rates at which Penta1 and Penta3 were administered was correlated (correlation between Penta1 and Penta3 = 0.75). The resulting median and 95% uncertainty intervals in the estimated proportion vaccinated over age in months are displayed in S12–S14 Figs and show good model fit to the raw DHS data.

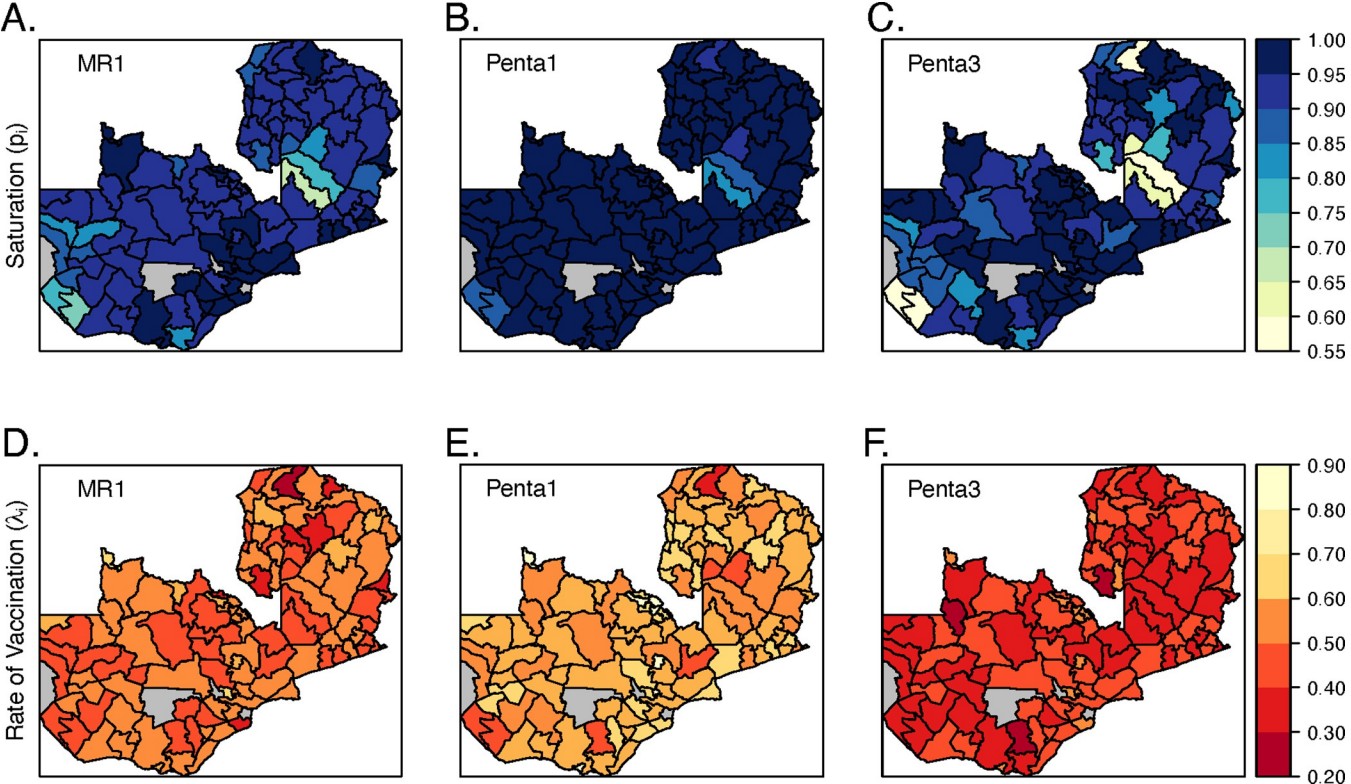

**Fig 2.** District-level (112 / 116 districts) median parameter estimates of the lifetime probability of being vaccinated via routine vaccination through 36 months of age (saturation parameter) (A-C), and monthly rate of receiving routine vaccination (D-F) for MR1 (A & F), Penta1 (B & E), and Penta 3 (C & F) vaccine doses. There are no parameter estimates for four districts colored in grey due to the lack of any DHS sampling clusters in these districts. Shapefile available CC BY 4.0 license via https://data.grid3.org/datasets/GRID3::nsdi-zambia-administrative-boundaries-districts-2022-published-by-grid3/about.

### National reduction in the pandemic year 2020 vaccination rate (compared to pre-pandemic years 2018 & 2019)

For all three vaccines, we found an increase in the proportion of the annual birth cohort vaccinated in the month of June (2018/2019 compared to 2020) because of Zambia's Child Health Week (Fig 3). The estimated rate of disruptions in routine vaccination during the COVID-19 pandemic varied between MR1, Penta1, and Penta3. Routine vaccination with MR1 in 2020 was lower from January through May compared to in 2018/2019; however, the Child Health Week in June helped to catch-up some missed children. There was a higher proportion of routine vaccination with Penta1 and Penta3 in September and October in pandemic year 2020 compared to 2018/2019. We noted a slight catch-up of missed children in June 2020 for Penta3 vaccination, similar MR1. The mean percent reduction in routine vaccination in the 2020 estimates from the 2018/2019 estimates for MR1 ranged from 15.66% to -6.04% across months (i.e., an increase by 6.04% in June 2020 compared to June 2018/2019); the mean percent reduction across months was 6.29% (95% uncertainty interval, 5.33% - 7.23%). The mean percent reduction in routine vaccination in the 2020 estimates from the 2018/2019 estimates for Penta1 ranged from 12.31% to -6.61% across months; the mean percent reduction across months was 3.73% (95% uncertainty interval, 2.79% - 4.69%). The mean percent reduction in routine vaccination in the 2020 estimates from the 2018/2019 estimates for Penta3 ranged from 12.14% to -6.88% across months; the mean percent reduction across months was 2.65% (95% uncertainty interval, 1.67% - 3.64%).

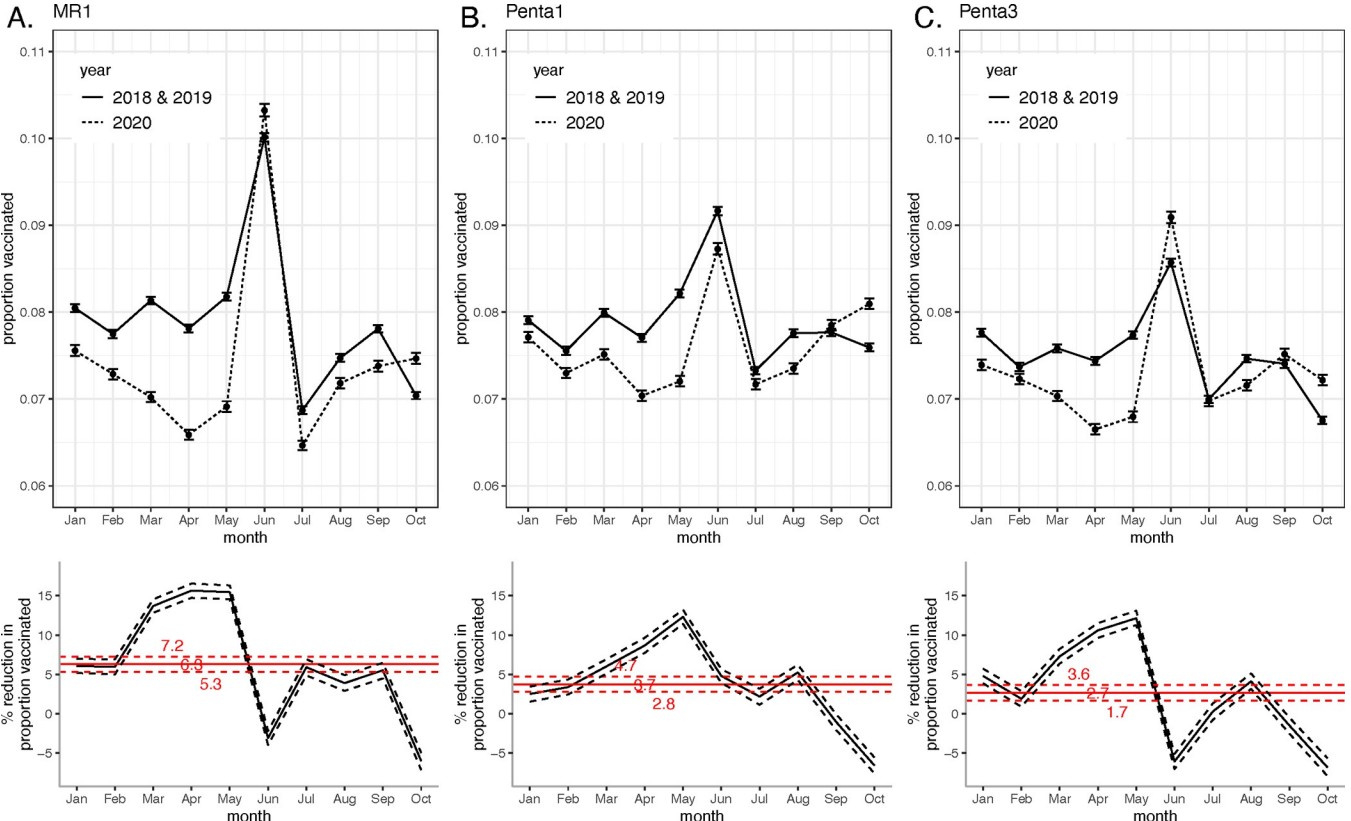

**Fig 3.** Disruption to routine vaccination for MR1 (A), Penta1 (B), Penta3 (C) based on administrative vaccination data. Top row is the proportion of the birth cohort vaccinated January to October in years 2018/2019 and 2020 (mean and 95% uncertainty intervals represented by points and error bars). Bottom row is the percent reduction in proportion vaccinated each month (black lines) and mean and 95% uncertainty intervals across months (red lines).

## District-specific number of children missed by vaccination

The number of additional children missed by vaccination in pandemic year 2020 was minimal compared to the total number missed in our pre-pandemic years 2018/2019. In the pre-pandemic years of 2018/2019, we estimate that 225,114.27 (95% CI 148,588.93–354,188.33) children missed MR1, 143,342.54 (95% CI 106,127.06–230,346.80) missed Penta1, and 268,572.48 (95% CI 179,582.99–419,802.18) missed Penta3 (Fig 4A–4C, S15 Fig). Given the median number of missed vaccinations in a non-disruption year, the percent increase in the number of doses missed in 2020 was largest for MR1 (2.80%) followed by Penta1 (2.13%) and Penta3 (1.19%). An additional 6,305.43 (95% CI 5,250.25–7,273.84) children missed MR1 after 10 months of disruption compared to the median number vaccinations missed in a non-disruption year (Fig 4A). Fewer additional children missed Penta1 and Penta3 vaccinations compared to the median number of vaccinations missed in a non-disruption year (Penta1 3,052.22 (95% CI 2,288.11–3,918.34), Penta3 3,198.74 (95% CI 1,994.91–4,302.01)) (Fig 4B and 4C).

There was large variation in the number of unvaccinated children across districts (Fig 4D–4F). For MR1 and Penta 3, Lusaka and Luangwa Districts (both located in Lusaka Province) had the highest and lowest number of children missed in a district, respectively. For Penta 1, Lusaka and Milenge Districts had the highest and lowest number of children missed in a district, respectively In Lusaka District, an estimated 25,352.04 children did not receive MR1, 16,895.25 children did not receive Penta1, and 32,405.49 children did not receive Penta3. In Luangwa District, an estimated 336.00 children did not receive MR1 and 313.15 children did not receive Penta3. In

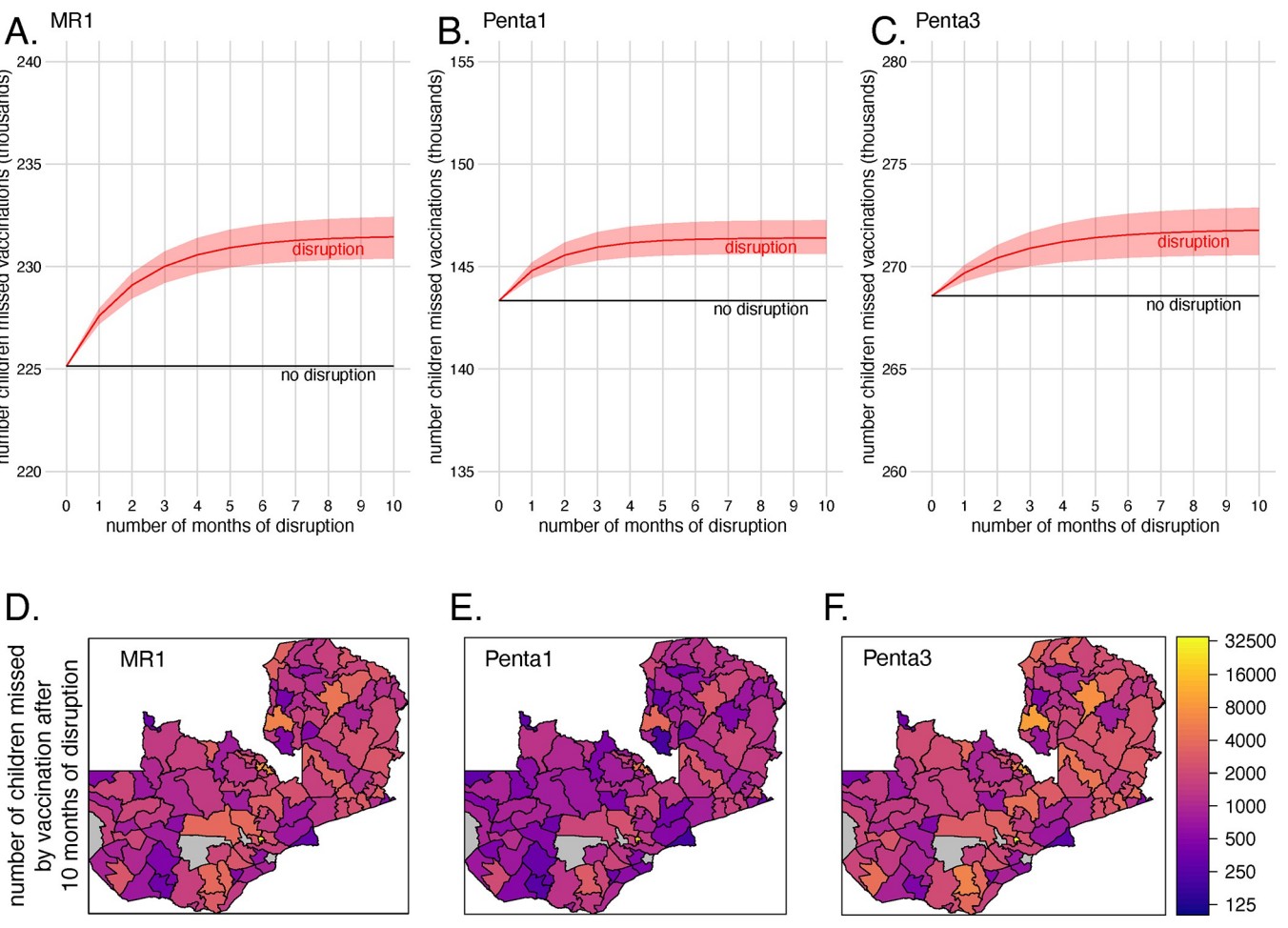

**Fig 4. Estimated number of children between the ages of 0 and 36 months missed by vaccination.** The national estimated cumulative median number of children missed by vaccination for MR1 (A), Penta1 (B), and Penta3 (C) by month of disruption (red line) across the range of disruption rates (red ribbon) based on the median number of children missed in reference year (black line). Broken down by district (112 / 116 districts) is the median number of children missed by MR1 (D), Penta1 (E), and Penta3 (F) vaccination after 10 months of disruption. Shapefile available CC BY 4.0 license via https://data.grid3.org/datasets/GRID3::nsdi-zambia-administrative-boundaries-districts-2022-published-by-grid3/about.

Milenge District an estimated 216.58 children did not receive Penta1. There were also a cluster of districts in Copperbelt Province that had a high number of unvaccinated children.

## District-specific measles outbreak risk

We also evaluated the impact of the COVID-19 pandemic disruptions to routine immunization services on the risk of measles outbreaks for each district over the course of the pandemic prior to the national MR vaccination campaign in November 2020. We found minimal impact on outbreak risk because of pandemic-year disruptions in routine vaccination or delay in fall 2020 MR vaccination campaign. Over the course of 10 months, measles $R_e$ increased on average across the districts by 0.05% (range 0.02% - 0.18%) (Fig 5). There was little to no change in $R_e$ during the two-month delay in conduction the MR campaign in all districts (Fig 5, S9 Fig).

## Discussion

We performed a detailed analysis to estimate changes in the national routine immunization program in Zambia during the pandemic year 2020, and its impact on the number of missed

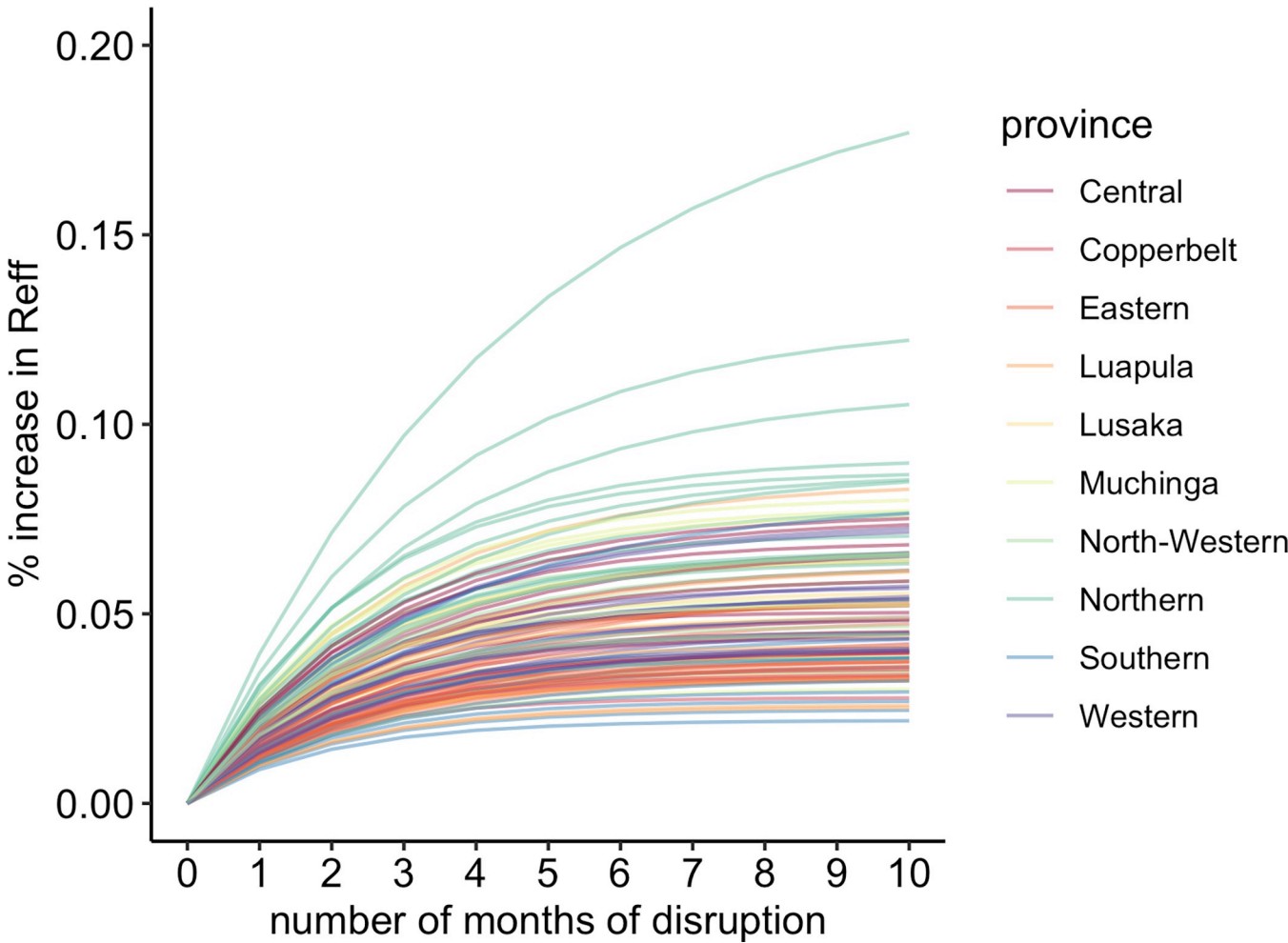

**Fig 5. Percent increase in $R_{eff}$ by month of disruption.** Each line represents a different district (112 / 116 districts), the color represents the province that each district is located.

children (and risk of measles outbreak) at the district level (administrative level 2). The methodological framework developed for this analysis can be used to estimate COVID-19 pandemic disruptions and its impact at the subnational scale in other countries or for other vaccines. Compared to pre-pandemic years 2018/2019, this study estimated minimal pandemic-related disruptions to childhood MR1, Penta1, and Penta3 vaccination in Zambia in 2020.

We estimated that the average percent reduction in proportion vaccinated between January and October 2022 was 6.29%, 3.73%, and 2.65% for MR1, Penta1, and Penta3 respectively. This is equivalent to an absolute reduction in the monthly proportion vaccinated by 0.005, 0.003, and 0.002. This translation to absolute reduction is in line with estimates generated by Causey et al. 2021, who estimated 0.5% (95% uncertainty intervals 0.3–0.6) and 0.2% (95% uncertainty intervals 0.1–0.5) reduction for MR1 and Penta3 respectively in Zambia in 2020 [2]. Among the 94 countries for which 2020 changes in routine vaccination were estimated by Causey et al. 2021, Zambia is in the top 10% of countries with minimal disruption (i.e., <0.5% decrease in 2020 MR1 coverage) [2].

Continued routine and catch-up immunization services during the pandemic have shown to be a net benefit in modelling studies [3]. Regardless, there remain concerns about the potential for SARS-CoV-2 transmission during routine or campaign vaccination activities because

of interactions with healthcare workers or other individuals seeking services. Zambia's Child Health Weeks conducted in June and November of 2020 took place despite the pandemic. The purpose of Zambia's Child Health Week, held biannually, is to reach eligible children who had not yet received their routine vaccines. This purpose was critically fulfilled during the pandemic year 2020 when the June Child Health Week resulted in a greater increase in the number of vaccinated children from the pre-pandemic years 2018/2019 for MR1 and Penta3. The MR vaccination campaign that was delayed two months was instituted during Zambia's second yearly Child Health Week in November 2020. This analysis demonstrates the benefits of continuing with routine immunization services during the pandemic and using catch-up vaccination activities to vaccinate those children who may have missed due to COVID-19 pandemic disruptions.

Ongoing collaborations and established research programs on measles and rubella in Zambia allowed a timely assessment of the impact of COVID-19 disruptions on routine immunization services and the impact of delaying a MR vaccination campaign for two months. For example, rich measles serological data collected in 2016 from a national serosurvey was used to set a baseline R effective for assessing changes in measles outbreak risk over months of disruption [20]. We identified minimal increases in risk of a measles outbreak due to postponing the MR vaccination campaign by two months; over 10 months measles $R_e$ increased by 0.05% on average across districts. R effective is the average number of people a typical with disease will go onto infect. It is an imprecise summary measure that hides variation and tends to have large uncertainty bounds [24, 25]. Assuming R effective is 1.00, a 0.05% increase means R effective is now 1.005; this is a minimal change relative to the variation in these estimates. In the end, the MR campaign was indeed delayed with no outbreaks reported over the course of those two months.

This study is subject to several limitations. The first is our assumption that differences in routine vaccination rates in 2020, compared to 2018/2019, is attributable to COVID-19 related disruptions. Given the stability of MR1 and Penta1 vaccination coverage leading up the pandemic year 2020, this assumption may hold. However, Penta3 was slightly declining in the 2018–2019 meaning that the changes we saw in 2020 to Penta3 vaccination may not be due to the disruptions from the pandemic. We surprisingly saw the smallest disruptions to Penta3, compared to Penta1 and MR1 vaccinations, rather than the expected largest disruptions in Penta3 if there was indeed an additive factor of decreasing pattern to the pandemic disruptions. Regardless, there is the potential for time-varying covariates, unrelated to the pandemic, that we did not consider within the analysis that could have explained the slight reductions in MR1, Penta1, and Penta3. Second, we focused on the potential of the COVID-19 pandemic to disrupt vaccination programs negatively. There are other potential ways the pandemic can impact the burden of vaccine-preventable diseases. For example, non-pharmaceutical interventions, such as movement restrictions (either personal or state enforced), can reduce transmission of directly transmitted infectious pathogens. Minimal disruption of routine vaccination programs coupled with decreased movement and transmission of vaccine-preventable diseases can inadvertently lead to local eliminations with minimal additional risk of resurgence. However, decreased circulating viruses can also create a short-term illusion of control without considering the potential risk of increasing susceptible populations [26, 27]. The third limitation is that the administrative vaccination data was not sufficient to estimate subnational disruptions to routine vaccinations; rather, we estimated a national rate of reduction with uncertainty based on district level data. Variation in the number of unvaccinated children across districts was driven by district-specific size of birth cohorts and pre-pandemic district-specific vaccination coverage and did not include the heterogeneities in disruptions to routine services over space. As a result, the variation in the number of unvaccinated children is likely

smaller than truly exists and should be interpreted through this lens. Fourth, our estimate of district-specific change in measles outbreak risk could be averaging across potential within-district heterogeneities in susceptibility [28]. Lastly, the hierarchical seroprevalence model was not suitable to generate districts' predicted risk of a measles outbreak during the pandemic, but simply the change in risk. Using our hierarchical model that includes district-specific random intercepts to predict seroprevalence profiles in 2020 would require a strong and unlikely assumption that the underlying district-specific impact on seroprevalence is constant from 2016 to 2020. A key area of future work is to build models that can reliably extrapolate seroprevalence to other years from rich population-based cross-sectional serological data. As suggested by this analysis, this may require individual level data on mechanisms of seroconversion (i.e., history of vaccination or measles infection) linked to the serum samples.

Since this analysis was completed, there has been additional waves of COVID-19 cases in Zambia. The implications of the pandemic on Zambia's childhood vaccination program are not fully realized. For example, the return wild poliovirus in Malawi and Mozambique in 2022 after many years of elimination, highlights the vulnerability of vaccination programs to disruptions to routine vaccination [29]. Further analysis is needed to evaluate the ongoing disruptions and understand potential subdistrict variations in the impact of the pandemic on childhood vaccinations.

Published literature has shown that the COVID-19 pandemic has resulted in disruptions to health systems including immunization programs worldwide, although lots of variability across countries [2, 4, 6, 8]. In this manuscript, we presented an analysis conducted early in the pandemic (mid-2020) in Zambia to estimate changes in the national routine immunization program in during the pandemic year 2020, and its impact on the number of children with missed vaccinations and measles outbreak risk at the district level. We estimated minimal pandemic-related disruptions to childhood MR1, Penta1, and Penta3 vaccination in Zambia in 2020, compared to pre-pandemic years 2018/2019. We found that continued supplemental immunization activities (e.g., child health week) were important to catch-up children who had missed their routine vaccination, and should continue to be prioritized, albeit safely. Lastly, this work highlights the utility of rich immunity profiles from serological data to evaluate changes in measles outbreak risk.

## Supporting information

**S1 Text. Estimating pre-pandemic routine vaccination coverage for MR1, Penta1, and Penta 3.**
(PDF)

**S2 Text. Estimating measles susceptibility by age (4 to 49 years old) using a model fit to serological data.**
(PDF)

**S3 Text. Estimating R effective.**
(PDF)

**S1 Fig. National level time trends in MR1, Penta1, and Penta3 vaccination coverage in Zambia 1980–2019.**
(PDF)

**S2 Fig. Impact of estimated parameters lambda and rho on the proportion vaccinated over age.**
(PDF)

**S3 Fig. District-specific administrative MR1 coverage estimates by month in years 2018, 2019, and 2020.**
(PDF)

**S4 Fig. District-specific administrative Penta1 coverage estimates by month in years 2018, 2019, and 2020.**
(PDF)

**S5 Fig. District-specific administrative Penta3 coverage estimates by month in years 2018, 2019, and 2020.**
(PDF)

**S6 Fig. Raw administrative vaccination data.**
(PDF)

**S7 Fig. Results of leave out district analysis.**
(PDF)

**S8 Fig. Results of leave out age analysis.**
(PDF)

**S9 Fig. Sensitivity analysis of percent increase in Reff per month of disruption based on different starting estimates of proportion susceptible across ages 4 to 49 years old.**
(PDF)

**S10 Fig. Comparison of estimated immunity by age (9–36 months) using seroprevalence data and 2018 Zambia DHS data.**
(PDF)

**S11 Fig. District-level estimates of the average age of vaccination among those that receive vaccination for MR1, Penta1, and Penta3.**
(PDF)

**S12 Fig. District-level MR1 baseline routine proportion vaccinated over age in months.**
(PDF)

**S13 Fig. District-level Penta1 baseline routine proportion vaccinated over age in months.**
(PDF)

**S14 Fig. District-level Penta3 baseline routine proportion vaccinated over age in months.**
(PDF)

**S15 Fig. Estimated number of children missed by vaccination of MR1, Penta1, and Penta3.**
(PDF)

## Author Contributions

**Conceptualization:** Amy K. Winter, Saki Takahashi, Kyla Hayford, Francis D. Mwansa, Nyambe Sinyange, William J. Moss, Simon Mutembo.

**Data curation:** Amy K. Winter, Saki Takahashi, Andrea C. Carcelen, Wilbroad Mutale, Francis D. Mwansa, Nyambe Sinyange, David Ngula, Simon Mutembo.

**Funding acquisition:** Kyla Hayford, William J. Moss.

**Investigation:** Wilbroad Mutale, Francis D. Mwansa, Nyambe Sinyange, David Ngula, William J. Moss, Simon Mutembo.

**Methodology:** Amy K. Winter, Saki Takahashi.

**Project administration:** William J. Moss, Simon Mutembo.

**Visualization:** Amy K. Winter, Saki Takahashi.

**Writing – original draft:** Amy K. Winter, Saki Takahashi, Andrea C. Carcelen, Simon Mutembo.

**Writing – review & editing:** Amy K. Winter, Saki Takahashi, Andrea C. Carcelen, Kyla Hayford, Wilbroad Mutale, Francis D. Mwansa, Nyambe Sinyange, David Ngula, William J. Moss, Simon Mutembo.

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
