## [Decision Letter · Decision Letter 0]

6 Oct 2022

PGPH-D-22-00439

An evaluation of the early impact of the COVID-19 pandemic on Zambia's routine immunization program

Dear Dr. Winter,

Thank you for submitting your manuscript to PLOS Global Public Health. After careful consideration, we feel that it has merit but does not fully meet PLOS Global Public Health’s publication criteria as it currently stands. Therefore, we invite you to submit a revised version of the manuscript that addresses the points raised during the review process.

The referees have raised a number of points that I have appended below and also in the attached file. If the paper can be substantially revised to take account of these comments I would be happy to reconsider it for publication.

We look forward to receiving your revised manuscript.

Kind regards,

Jong-Hoon Kim, Ph.D.

Academic Editor

Journal Requirements:

1. Please include a complete copy of PLOS’ questionnaire on inclusivity in global research in your revised manuscript. Our policy for research in this area aims to improve transparency in the reporting of research performed outside of researchers’ own country or community. The policy applies to researchers who have travelled to a different country to conduct research, research with Indigenous populations or their lands, and research on cultural artefacts. The questionnaire can also be requested at the journal’s discretion for any other submissions, even if these conditions are not met.  Please find more information on the policy and a link to download a blank copy of the questionnaire here: https://journals.plos.org/plosone/s/best-practices-in-research-reporting. Please upload a completed version of your questionnaire as Supporting Information when you resubmit your manuscript.

2. We ask that a manuscript source file is provided at Revision. Please upload your manuscript file as a .doc, .docx, .rtf or .tex.

3. Please provide separate figure files in .tif or .eps format only and remove any figures embedded in your manuscript file. Please also ensure that all files are under our size limit of 10MB.

4.We have noticed that you have uploaded Supporting Information files, but you have not included a list of legends. Please add a full list of legends for your Supporting Information files after the references list. 

5. In the online submission form, you indicated that "Data can be made available upon request". All PLOS journals now require all data underlying the findings described in their manuscript to be freely available to other researchers, either 1. In a public repository, 2. Within the manuscript itself, or 3. Uploaded as supplementary information.

Additional Editor Comments (if provided):

Reviewers' comments:

Reviewer's Responses to Questions

**Comments to the Author**

1. Does this manuscript meet PLOS Global Public Health’s publication criteria? Is the manuscript technically sound, and do the data support the conclusions? The manuscript must describe methodologically and ethically rigorous research with conclusions that are appropriately drawn based on the data presented.

Reviewer #1: Yes

Reviewer #2: Yes

2. Has the statistical analysis been performed appropriately and rigorously?

Reviewer #1: I don't know

Reviewer #2: Yes

3. Have the authors made all data underlying the findings in their manuscript fully available (please refer to the Data Availability Statement at the start of the manuscript PDF file)?

Reviewer #1: No

Reviewer #2: Yes

4. Is the manuscript presented in an intelligible fashion and written in standard English?

Reviewer #1: Yes

Reviewer #2: Yes

5. Review Comments to the Author

Reviewer #1: Thank you for the opportunity to review this study. Overall, it offered important insights into how the ongoing COVID-19 pandemic may have affected routine immunization in Zambia, with a primary focus on potential disruptions in 2020.

There a few areas of the current manuscript that could be strengthened, or at least contextualized more, before publication. They are as follows (in the attached review), organized by major and minor comments; note that exact page and paragraph attribution may not be 100% accurate in the absence of included page and line numbers.

Reviewer #2: Major comments

Regarding “As it is difficult to establish a causal link between the pandemic and vaccination programs, here we compared pre-pandemic and pandemic vaccination program performance and attributed the difference to the pandemic” and “The disruption rate per month was estimated as the percent reduction between the 2018 and 2020 in monthly estimates”:

Were immunization rates already increasing or decreasing prior to the pandemic? The authors can use data from previous DHS surveys to assess this

- If stable before the pandemic, that’s ideal and validates an assumption of this study

- If already decreasing before the pandemic, need to argue for why observed decreases are the effect of the pandemic and not due to exogenous causes.

- If already increasing before the pandemic, an observation of no change between 2018 and 2020 would be a potentially important finding, indicating that the pandemic potentially inhibited further increases. In this case, I’d recommend a difference-in-differences analysis to quantify the effect size.

Generally needs better discussion of potential time-varying confounders.

--

The authors assessed change over time using the number of doses administered in 2018 and 2020, respectively. By using this metric as a proxy for immunization rates, it is implicitly assumed that the population denominator is constant over time. This seems unlikely to be true given Zambia’s generally large increase in population in recent years, partially driven by improvements in child survival which would increase the size of the child population. If the child population is indeed increasing and not captured in this analysis, this would lead to upward bias in change in immunization rates over time

- Do WorldPop data (“the size of the population in the district i estimated by aggregating WorldPop population estimates in 10x10 km grid cells over district”) have enough time granularity to capture this effect?

- If not, I encourage the authors to discuss this potential bias in the Discussion section or attempt to adjust for it quantitatively

--

Minor comments

Regarding “the lower range of the disruption rate per month was estimated as the percent difference between the 2.5th quantile of the 2018 month estimates and the 97.5th quantile of the 2020 estimates”:

This seems backwards to me; 2.5th quantile in 2018 should be compared to 2.5th quantile in 2020. Or if my comment is incorrect, I would like to see a better explanation for the method of quantifying uncertainty.

--

Regarding “We found minor disruptions to routine administration of measles-rubella and pentavalent vaccines in 2020. This was in part due to Zambia’s Child Health Week held in June of 2020 which helped to reach children missed during the first six months of the year”:

What exactly was due to the Child Health Week (CHW)? This reads as “disruptions” were due to CHW, but I think the authors mean the minor-ness of the disruption was due to CHW. Please re-word.

--

The analysis assumes that the sampled populations in the pre- and post-pandemic periods are not systematically different. The authors should argue for why they think this assumption holds in spite of many societal-level factors changing in an unprecedented way during the study period.

--

The section assessing measles risk is a valuable addition to the study and appears well-conducted.

--

Regarding "However, it is difficult to compare the number of children who received vaccination in November of 2020 to November of 2018 because the target population is inconsistent between the two years (2018 9-23 months, 2020 9-59 months); therefore, the month of November was excluded from the disruptions estimates between November 2020 and March 2021 for MR1."

- Discarding data is a drastic step and I do not fully understand why it was necessary. 9-59 months is a superset of 9-23 months, so the two periods seem compatible me. Please elaborate.

- Please provide a sensitivity analysis showing the results with and without the discarded November data.

6. PLOS authors have the option to publish the peer review history of their article (what does this mean?). If published, this will include your full peer review and any attached files.

**Do you want your identity to be public for this peer review?** For information about this choice, including consent withdrawal, please see our Privacy Policy.

Reviewer #1: No

Reviewer #2: No

---

## [Decision Letter · Decision Letter 1]

3 Apr 2023

An evaluation of the early impact of the COVID-19 pandemic on Zambia's routine immunization program

PGPH-D-22-00439R1

Dear Dr Winter,

We are pleased to inform you that your manuscript 'An evaluation of the early impact of the COVID-19 pandemic on Zambia's routine immunization program' has been provisionally accepted for publication in PLOS Global Public Health.

Best regards,

Jong-Hoon Kim, Ph.D.

Academic Editor

Reviewer Comments (if any, and for reference):

Reviewer's Responses to Questions

**Comments to the Author**

1. If the authors have adequately addressed your comments raised in a previous round of review and you feel that this manuscript is now acceptable for publication, you may indicate that here to bypass the “Comments to the Author” section, enter your conflict of interest statement in the “Confidential to Editor” section, and submit your "Accept" recommendation.

Reviewer #1: All comments have been addressed

Reviewer #2: All comments have been addressed

2. Does this manuscript meet PLOS Global Public Health’s publication criteria? Is the manuscript technically sound, and do the data support the conclusions? The manuscript must describe methodologically and ethically rigorous research with conclusions that are appropriately drawn based on the data presented.

Reviewer #1: Yes

Reviewer #2: Yes

3. Has the statistical analysis been performed appropriately and rigorously?

Reviewer #1: I don't know

Reviewer #2: Yes

4. Have the authors made all data underlying the findings in their manuscript fully available (please refer to the Data Availability Statement at the start of the manuscript PDF file)?

Reviewer #1: Yes

Reviewer #2: Yes

5. Is the manuscript presented in an intelligible fashion and written in standard English?

Reviewer #1: Yes

Reviewer #2: Yes

6. Review Comments to the Author

Reviewer #1: The revised manuscript reflects substantial work by the authors and it is much improved. Well done!

Reviewer #2: The revised version addresses my comments and I have no further input. Nice work.

7. PLOS authors have the option to publish the peer review history of their article (what does this mean?). If published, this will include your full peer review and any attached files.

**Do you want your identity to be public for this peer review?** For information about this choice, including consent withdrawal, please see our Privacy Policy.

Reviewer #1: No

Reviewer #2: No
